# Cost-sensitive multi-kernel ELM based on reduced expectation kernel auto-encoder

**Liang Yixuan** [1,2]*

**1** School of Science, Xi 'an University of Technology, Xi'an, Shaanxi, P. R. China, **2** The University of Melbourne, Parkville, Victoria, Australia

* liangyx1202@163.com

## Abstract

ELM (Extreme learning machine) has drawn great attention due its high training speed and outstanding generalization performance. To solve the problem that the long training time of kernel ELM auto-encoder and the difficult setting of the weight of kernel function in the existing multi-kernel models, a multi-kernel cost-sensitive ELM method based on expectation kernel auto-encoder is proposed. Firstly, from the view of similarity, the reduced kernel auto-encoder is defined by randomly selecting the reference points from the input data; then, the reduced expectation kernel auto-encoder is designed according to the expectation kernel ELM, and the combination of random mapping and similarity mapping is realized. On this basis, two multi-kernel ELM models are designed, and the output of the classifier is converted into posterior probability. Finally, the cost-sensitive decision is realized based on the minimum risk criterion. The experimental results on the public and realistic datasets verify the effectiveness of the method.

**Data Availability Statement:** [1] Liang (2024). dataset.rar. figshare. Dataset. https://doi.org/10.6084/m9.figshare.27679113.v1 [2] https://figshare.com/articles/dataset/LungHist700_A_Dataset_of_Histological_Images_for_Deep_Learning_in_Pulmonary_Pathology/25459174?file=45206104.

## 1. Introduction

Extreme learning machine (ELM) [1] is a training algorithm for single-hidden-layer feedforward neural networks (SLFNs) that aims to minimize the training error while ensuring the minimum norm of output weights [2]. Its distinctive feature lies in the random initialization of input layer weights and hidden layer biases, which remain fixed throughout the training process.

Although ELM's randomly generated network parameters guarantee the infinite approximation capability of SLFNs [3], researches have shown that ELM requires a larger number of hidden nodes compared to traditional SLFNs to achieve similar performance [2,4]. Studies addressing this issue primarily focus on two ways. One is determining the optimal number of hidden nodes to construct a compact network structure [1]. Another way focuses setting effective hidden layer node parameters. The most common approaches employ optimization algorithms to search for effective input layer weights and hidden layer biases [5–8]. However, these methods require multiple ELM training iterations, resulting in high computational overhead, which contradicts ELM's design principle.

**Funding:** The author(s) received no specific funding for this work.

**Competing interests:** The authors have declared that no competing interests exist.

Kernel ELM (KELM) [1] obviates the need to set the number of hidden nodes and parameters, exhibiting improved stability and generalization ability compared to ELM. Reduced KELM (RKELM) [9] further enhances training efficiency by randomly selecting a subset of training samples as support vectors. The effectiveness of kernel methods stems from the ability of kernel functions to compute complex similarities between different samples [10]. Kärkkäinen [11] adopted Euclidean distance to measure the relationship between arbitrary training samples and selected reference points in ELM, demonstrating that similarity function-based hidden layer outputs ensure the infinite approximation capability of ELM. Therefore, from the perspective of similarity functions, it is possible to reinterpret the mechanism of kernel mapping, guiding the selection of kernel functions.

On the other hand, different kernel functions possess varying data mapping capabilities. To fully exploit the advantages of different kernel functions, ELM based on multiple kernel functions has been proposed. The most common multi-kernel model involves linearly combining multiple kernel functions [12,13]. However, it is difficult to predict the contribution of different kernel functions to classification performance, and determining appropriate weights for each kernel function to maximize the model's generalization performance remains a significant challenge in multi-kernel learning. While some researchers have proposed using optimization methods to determine the optimal weights [14], the training process incurs high computational costs. Consequently, designing multi-kernel fusion strategies remains an open problem.

Multilayer ELM (ML-ELM) [14] achieves deep representation of input features by stacking multiple ELM autoencoders (ELM-AE). Similar approaches include hierarchical ELM (H-ELM) [15], KELM-based autoencoders (KELM-AE) [16]. These methods construct deep network models by vertically stacking multiple autoencoders, enabling layer-by-layer representation of input features. In contrast, Wang [17] arranged multiple autoencoders horizontally, fusing the encoding results into a single vector. Overall, while deep learning based on ELM-AE has yielded fruitful results, horizontal expansion of networks based on ELM-AE remains an under-explored area, and research on autoencoders based on multiple kernels or similarity functions is still lacking. replaces traditional predefined kernel functions with multi-layer complex mappings of DNN [18].

Cost-sensitive learning has drawn great attention in both theoretical researches and engineering practices. Theoretical researches can be divided into two categories, direct cost-sensitive learning and indirect cost-sensitive learning [19]. The former methods intended to use cost information to design new variants based on the basic classifiers, while the later ones treated the traditional classification algorithms as black boxes and performed preprocessing on the training data or post-processing on the outcomes with the cost information [20]. For engineering practices, cost-sensitive learning has been widely used in imbalanced classification [1], biomedicine [21], fault diagnosis [22], etc. Traditional ELM-based cost-sensitive learning mainly depended on the original ELM model [1]. Therefore, exploration based on other new variants of ELM is always needed.

Based on the aforementioned analysis, this paper proposes two multi-kernel cost-sensitive ELM models based on expected kernel functions. Reinterpreting the significance of kernel functions from the perspective of similarity, we present a simplified kernel autoencoder model by randomly selecting a subset of samples from the input data as reference points. Subsequently, inspired by the theory of expected kernel ELM, we add a random mapping layer after the input layer, designing a simplified expected kernel autoencoder that effectively combines random mapping and similarity mapping. Finally, we define four similarity kernel functions and utilize the simplified expected kernel autoencoder to design two multi-kernel ELM models, converting the classifier output into posterior probabilities and implementing cost-sensitive decision-making based on the minimum risk criterion.

## 2. Related work

### 2.1 ELM and KELM

Let a training set consisting of $N$ training samples is denoted as $U = \{(\mathbf{x}_i, \mathbf{t}_i) | \mathbf{x}_i \in \mathbf{R}^d, \mathbf{t}_i \in \mathbf{R}^m\}$, where $\mathbf{x}_i = [x_{i1}, \cdots, x_{id}]^T$ is a set of $d$-dimensional input vectors, and the corresponding class vectors is $\mathbf{t}_i = [t_{i1}, \cdots, t_{im}]^T$, where $m$ is the number of classes. For a single-hidden-layer neural network with $L$ hidden nodes and $m$ output nodes, the output function of ELM can be represented as:

$$f_i(\mathbf{x}) = \sum_{j=1}^{L} \beta_{ij} h_j(\mathbf{w}_j, b_j, \mathbf{x}) = \mathbf{h}(\mathbf{w}, \mathbf{b}, \mathbf{x})\boldsymbol{\beta}_i, i = 1, 2, \cdots, m \tag{1}$$

Where $\beta_{ij}$ represents the output weights connecting the $j$-th hidden node to the $i$-th output node, and $\boldsymbol{\beta}_i = [\beta_{i1}, \beta_{i2}, \cdots \beta_{iL}]^T, i = 1, 2, \cdots, m$. $h_j(\mathbf{w}_j, b_j, \mathbf{x})$ is the output corresponding to the $j$-th hidden node for input sample $\mathbf{x}$, $\mathbf{w}_j$ is the randomly generated weights connecting input to the $j$-th hidden node, and $\mathbf{W} = [\mathbf{w}_1, \cdots, \mathbf{w}_L]^T$, $b_j$ is the bias for the $j$-th hidden node.

Let $\mathbf{h}(\mathbf{x}) = [h_1(\mathbf{w}_1, b_1, \mathbf{x}), \cdots, h_L(\mathbf{w}_L, b_L, \mathbf{x})]$, Eq (1) can be rewritten in matrix form as:

$$\mathbf{H}\boldsymbol{\beta} = \mathbf{T} \tag{2}$$

The solution of Eq (2) can be expressed in two forms:

$$\begin{cases} \boldsymbol{\beta} = \mathbf{H}^T(\dfrac{\mathbf{I}}{C} + \mathbf{H}\mathbf{H}^T)^{-1}\mathbf{T}, \ if : N << L \\ \boldsymbol{\beta} = (\dfrac{\mathbf{I}}{C} + \mathbf{H}^T\mathbf{H})^{-1}\mathbf{H}^T\mathbf{T}, \ if : N >> L \end{cases} \tag{3}$$

$C$ is the regularization parameter. When the form of $h_j(\cdot)$ is unknown, a kernel matrix can be defined for the ELM using the Mercer criterion:

$$\boldsymbol{\Omega} = \mathbf{H}\mathbf{H}^T, \boldsymbol{\Omega}_{i,j} = h(\mathbf{x}_i) \cdot h(\mathbf{x}_j) = K(\mathbf{x}_i, \mathbf{x}_j) \tag{4}$$

Replacing $\mathbf{H}\mathbf{H}^T$ with $\boldsymbol{\Omega}$ in Eq (4), the output function of KELM can be expressed as:

$$f(\mathbf{x}) = h(\mathbf{x})\mathbf{H}^T(\dfrac{\mathbf{I}}{C} + \boldsymbol{\Omega})^{-1}\mathbf{T} = \begin{bmatrix} K(\mathbf{x}_i, \mathbf{x}_1) \\ \vdots \\ K(\mathbf{x}_i, \mathbf{x}_N) \end{bmatrix} (\dfrac{\mathbf{I}}{C} + \boldsymbol{\Omega})^{-1}\mathbf{T} \tag{5}$$

ELM-AE [14] (as shown in Fig 1A) is essentially an ELM where the input and output are the same. Let $\boldsymbol{\Gamma}$ be the output weights of the autoencoder. The solution of ELM-AE in this case can be expressed as:

$$\begin{cases} \boldsymbol{\Gamma} = \mathbf{H}^T(\dfrac{\mathbf{I}}{C} + \mathbf{H}\mathbf{H}^T)^{-1}\mathbf{X}, \ if : N << L \\ \boldsymbol{\Gamma} = (\dfrac{\mathbf{I}}{C} + \mathbf{H}^T\mathbf{H})^{-1}\mathbf{H}^T\mathbf{X}, \ if : N >> L \end{cases} \tag{6}$$

Based on KELM, Wong [16] replaced the random mapping function in the autoencoder with a kernel function, proposed KELM-AE (as shown in Fig 3B), and its output layer

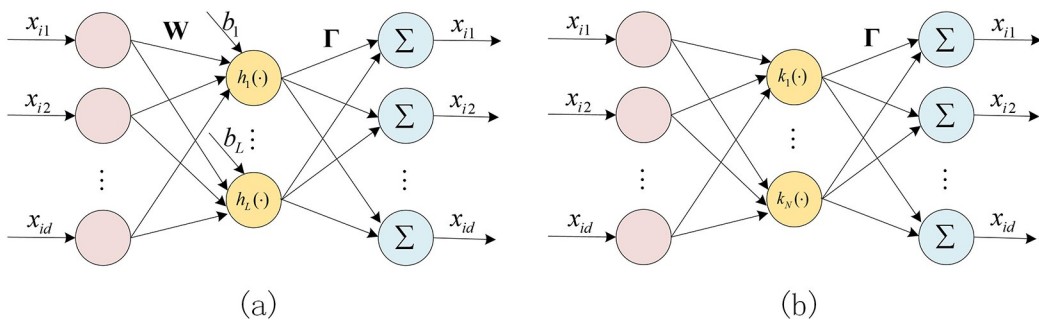

**Fig 1.** (a) ELM Autoencoder. (b) KELM Autoencoder.

encoding matrix can be represented as:

$$\mathbf{\Gamma} = (\frac{\mathbf{I}}{C} + \mathbf{\Omega})^{-1}\mathbf{X} \tag{7}$$

It has been demonstrated composite kernels could efficiently combines multi-level information and have excellent performance [12]. In [12], composite kernels is used for mapping contextual and spectral information,

$$K(\mathbf{x}_i, \mathbf{x}_j) = \mu K_1(\mathbf{x}_i, \mathbf{x}_j) + (1 - \mu)K_2(\mathbf{x}_i, \mathbf{x}_j) \tag{8}$$

Where $\mu$ is balance factor and $K_1(\mathbf{x}_i,\mathbf{x}_j)$, $K_2(\mathbf{x}_i,\mathbf{x}_j)$ are Radial basis function and polynomial function, respectively. The same methodology was adopted by [13]. However, the method above introduces one hyper-parameter, the balance factor, which is difficult to set. Therefore, [23] integrated multi-kernel matrix by adding each matrix immediately. Different from other articles that adopt traditional kernel functions, [18] redefined the kernel matrix as,

$$K(\mathbf{x}_i, \mathbf{x}_j) = \Phi_1(\mathbf{x}) \cdot \Phi_2(\mathbf{x}) \cdot \dots \cdot \Phi_n(\mathbf{x}) \tag{9}$$

Where $\Phi_i(\mathbf{x})$ is the hidden layer output of a neural network. Instead of integrating multi-kernel functions into one super kernel, [16] proposed ML-KELM by stacking multiple KELM-AEs, which does not need to tune the parameters for all layers as in ML-ELM. In ML-KELM, each encoder has one distinct kernel function.

## 2.2 Cost-sensitive learning based on ELM

ELM has attracted extensive attention from scholars in the field of cost-sensitive learning, and many cost-sensitive variations of ELM have been proposed.

For class-imbalanced problems, the most intuitive approach is to directly weight the samples according to their numbers [21]:

$$W_{ii} = \frac{Ny_{\min}}{Ny_i} \tag{10}$$

Where $W_{ii}$ is the squared error weight of $\boldsymbol{x}_i$, $Ny_{\min}$ is the number of samples in the smallest class, and $Ny_i$ is the number of samples in class $y_i$. The Cost-sensitive ELM model (CELM)

[24] utilizes misclassification cost information to weight the classification errors of samples:

$$\min \frac{1}{2}\|\boldsymbol{\beta}\|^2 + \frac{\lambda}{2}\sum_{i=1}^{N} c_i \boldsymbol{\varepsilon}_i^2 \tag{11}$$
$$s.t. \ \ \boldsymbol{h}(\boldsymbol{x}_i)\boldsymbol{\beta} = \boldsymbol{y}_i - \boldsymbol{\varepsilon}_i$$

Where $c_i$ is the misclassification cost of the $i$-th sample. Zhu[25] then uses the sum of misclassification costs for all classes in the cost matrix as weights:

$$k_i = \sum_{j=1}^{m} c_{ij} \tag{12}$$

When constructing a cost-sensitive ELM model, Zhang [26] takes into account both the class imbalance of the training data and misclassification costs, i.e:

$$\min \frac{1}{2}\|\boldsymbol{\beta}\|^2 + \frac{\lambda}{2} \cdot diag(\boldsymbol{B}) \cdot \sum_{i=1}^{N} \boldsymbol{\varepsilon}_i^2 \tag{13}$$
$$s.t. \ \ \boldsymbol{h}(\boldsymbol{x}_i)\boldsymbol{\beta} = \boldsymbol{y}_i - \boldsymbol{\varepsilon}_i, \ i = 1, 2, \cdots, N$$

Where $\boldsymbol{B}$ is a cost information vector determined by the class weight matrix and the misclassification cost matrix. Similarly, Fatemeh [27,28] proposed a hierarchical ELM (H-ELM) classification method, which consists of two successive stages, an unsupervised hierarchical feature representation and a supervised feature classification. In H-ELM, the weights of different classes were defined as follows:

$$W_{ii} = \frac{1}{p_i + (Ny_i - p_i) \times \frac{Ny_i}{\max(Ny_i)}}, \ p_i = \sum_i Ny_i - Ny_i \tag{14}$$

In addition, indirect cost-sensitive methods based on ELM mainly involve ensemble of multiple ELM classifiers [29], or adjusting the classifier output thresholds [22].

Although the above methods have achieved some results in specific application areas, the vast majority of cost-sensitive models mentioned above utilize the original ELM as the base classifier, with little research on multi-layered or parallel cost-sensitive ELM models. Additionally, the use of ensemble methods to estimate posterior probabilities can fully utilize cost matrix information, but requires training multiple classifiers. For example, Lu's method [30] requires training thirty classifiers each time, resulting in high time overhead when dealing with large training data. Therefore, further research is needed in cost-sensitive learning based on ELM.

## 3. Double hidden layer autoencoder based on simplified expected kernel function

In this section, a Reduced Kernel Extreme Learning Machine (RKELM) is first introduced, then a RKELM-based autoencoder (RKELM-AE) is proposed. Subsequently, a Reduced Expectation Kernel Autoencoder (REKELM-AE) is proposed by adding a random mapping layer after the input layer of RKELM-AE according to the theory of expected kernel.

### 3.1 RKELM-AE

In KELM, each training sample is measured for similarity with all other training samples [10]. For any training sample $\mathbf{x}_i$, KELM essentially creates a new feature for $\mathbf{x}_i$ by measuring its similarity with samples of different classes. Let $\mathbf{r}_i$ (reference point) be the reference point used for

measuring similarity with training samples, and let $\mathbf{R} = \{\mathbf{r}_i\}_{i=1}^m$ be the collection of all reference points. In KELM, $\mathbf{R}$ represents the training set, and each reference point in $\mathbf{R}$ corresponds to each dimension of the kernel space. However, when two reference points in $\mathbf{R}$ are close, the similarity between the training samples and these two reference points is approximately the same. Therefore, the features corresponding to these two reference points in the kernel space after mapping will be similar. Hence, when the number of training samples is large or the distribution is compact, it is unnecessary to select all samples as reference points. Kärkkäinen [11] have validated the effectiveness of randomly selecting a subset of samples from the training data as reference points for KELM from different perspectives.

Based on the analysis above, the proposed RKELM-AE, building upon KELM, randomly selects a subset of $\bar{N}$ samples $\mathbf{R} = \{\mathbf{r}_i\}_{i=1\bar{N}}$ from the $N$ training samples as reference points to map the input data for similarity. Similar to ELM-AE, the output matrix of RKELM-AE can be represented as:

$$\bar{\mathbf{\Omega}}\mathbf{\Gamma} = \mathbf{X} \tag{15}$$

Where $\bar{\mathbf{\Omega}}$ is equal to $N \times \bar{N}$, and $\bar{\mathbf{\Omega}}_{i,j} = K(\mathbf{x}_i, \mathbf{r}_j)$, $i = 1, 2, \cdots, N, j = 1, 2, \cdots, \bar{N}$, each row represents the similarity of a sample to $\bar{N}$ reference points. The output matrix can be represented as:

$$\mathbf{\Gamma} = (\frac{\mathbf{I}}{C} + \bar{\mathbf{\Omega}}T\bar{\mathbf{\Omega}})^{-1}\bar{\mathbf{\Omega}}T\mathbf{X} \tag{16}$$

In Eq (13), the computational complexity of KELM-AE is $O(N^3)$. When the number of samples is large, solving for the output matrix will still take significant time. However, in RKELM-AE, selecting only $\bar{N} \times N$ samples can keep the reconstruction error within a small range. Therefore, the computational complexity of RKELM-AE is much smaller than that of KELM-AE.

## 3.2 REKELM-AE

The main feature of ELM is the random generation of hidden layer weights. However, in ELM models based on kernel functions, the mapping of the hidden layer is deterministic or semi-deterministic (such as RKELM), and ELM also can learn effective information based on this stable mapping. By combining these two models, one can simultaneously leverage the advantages and disadvantages of both mapping behaviors. To achieve this, Zhang [10] proposed the concept of the Expectation Kernel (EK) to study the relationship between random mapping and kernel functions. The definition of linear EK is:

$$\begin{aligned} K_e(\mathbf{x}_i, \mathbf{x}_j) &= E[h(\mathbf{v}^T\mathbf{x}_i)h(\mathbf{v}^T\mathbf{x}_j)] \\ &= \int_{R^{d+1}} p(\mathbf{v})h(\mathbf{v}^T\mathbf{x}_i)h(\mathbf{v}^T\mathbf{x}_j)d\mathbf{v} \end{aligned} \tag{17}$$

Where $\mathbf{v}^T\mathbf{x}_i = \mathbf{w}^T\mathbf{x}_i + b$. $p(\mathbf{v})$ is the probability distribution function of weights and biases. Sampling $\mathbf{v}$ according to $p(\mathbf{v})$, the linear EK in Eq (20) can be approximated as:

$$\begin{aligned} K_e(\mathbf{x}_i, \mathbf{x}_j) &= \frac{1}{N}\sum_{k=1}^N h(\mathbf{v}_k^T\mathbf{x}_i)h(\mathbf{v}_k^T\mathbf{x}_j) \\ &= \frac{1}{N}\langle h(\mathbf{V}^T\mathbf{x}_i), h(\mathbf{V}^T\mathbf{x}_j)\rangle \end{aligned} \tag{18}$$

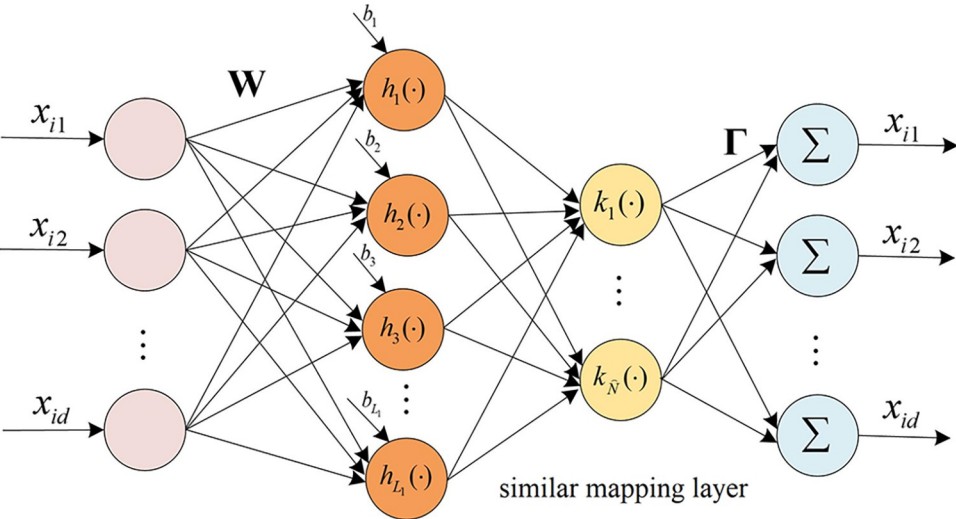

**Fig 2. REKELM-AE architecture.**

Where $\mathbf{V} = [\mathbf{v}_1, \cdots, \mathbf{v}_L]$ is the weight matrix randomly sampled according to $p(\mathbf{v})$. However, the linear EK and the standard ELM mapping process are essentially equivalent, both can be viewed as randomly initializing the hidden layer weights and mapping the input randomly. Therefore, Zhang built upon the linear EK and defined a non-linear EK using traditional kernel functions $K(\cdot)$:

$$K_{ne}(\mathbf{x}_i, \mathbf{x}_j) = K(h(\mathbf{V}^T\mathbf{x}_i), h(\mathbf{V}^T\mathbf{x}_j)) \tag{19}$$

According to Eq (22), the non-linear EK can be divided into two independent processes: randomly mapping the original data, and using the results of the random mapping for kernel mapping.

Following the concept of EK, a random mapping layer consisting of $L_1$ hidden layer nodes is added after the input layer of RKELM-AE, allowing for high-dimensional random mapping of the original data. The simplified structure of the Expected Kernel Autoencoder REKELM-AE is defined as shown in Fig 2.

REKELM-AE can be seen as a combination of traditional ELM and RKELM: the first part utilizes the random hidden layers of ELM to map the input to ELM space, while the second part uses the simplified kernel ELM to compute the similarity of the random mapping results.

For the training samples, the mapping result after the first hidden layer of REKELM-AE is:

$$\mathbf{H}_{N \times L_1} = \begin{bmatrix} h(\mathbf{V}^T\mathbf{x}_1)^T \\ h(\mathbf{V}^T\mathbf{x}_2)^T \\ \vdots \\ h(\mathbf{V}^T\mathbf{x}_N)^T \end{bmatrix} \tag{20}$$

Then, randomly select $\bar{N}$ reference points $\mathbf{R} = \{\mathbf{r}_i\}_{i=1\bar{N}}$ from the $N$ rows of samples $\mathbf{H}_{N \times L_1}$, and calculate the simplified matrix:

$$\bar{\mathbf{\Omega}}_{i,j} = K(h(\mathbf{V}^T\mathbf{x}_i), \mathbf{r}_j), i = 1, 2, \cdots N, j = 1, 2, \cdots, \bar{N} \tag{21}$$

Finally, the output matrix $\mathbf{\Gamma}$ can be calculated based on Eq (19). The training procedure of REKELM-AE is presented as follows,

```
Algorithm 1. pseudocode for REKELM-AE.
Input X, L₁, N̂, p(v), h(·), K(·), C
1. Randomly generate W,b according to p(v)
2. Generate random mapping matrix H using W,b
3. Select N̂ reference points from H randomly
4. Calculate the simplified matrix Ω̂ according to Eq (21)
5. Calculate output matrix Γ according to Eq (16)
Output Γ
```

It should be noted that although REKELM-AE introduces a random mapping process compared to RKELM-AE, the complexity of this process is only related to the number of selected reference points after simplification and is independent of the dimensionality of the samples after dimensionality augmentation. Therefore, REKELM-AE retains the efficiency of RKELM-AE.

## 4. Cost-sensitive ELM with multi-kernel autoencoder

This section first proposes two classification models based on REKELM-AE, namely multi-kernel parallel ELM (MKP-ELM) and multi-kernel residual ELM (MKR-ELM). Then, the cost-sensitivity models are realized according to the minimum risk decision theory.

### 4.1 MKP-ELM and MKR-ELM

Most existing AE-based neural network models achieve deep representation of inputs by vertically stacking multiple AEs. However, this layer-by-layer learning strategy accumulates reconstruction errors in the process of feature reconstruction, leading to the learned features cannot represent the original input information well [16]. On the other hand, different kernel functions have different expressive capabilities for different data. By combining multiple kernel functions for learning, we can leverage the advantages of different kernel functions, obtaining better flexibility and adaptability. Based on the REKELM-AE proposed in the third section, this paper presents two multi-kernel autoencoder-based ELM models, multi-kernel parallel ELM (MKP-ELM) and multi-kernel residual ELM (MKR-ELM), as shown in Fig 3.

Both of the two models consist of two separate processes: feature extraction based on multi-kernel parallel autoencoders, and ELM classification based on the extracted features. For MKP-ELM, firstly, given $k$ different kernel functions, each corresponding to a REKELM-AE, calculate the output weight $\mathbf{\Gamma}^{(i)}, i = 1,2,\cdots,k$ of the corresponding autoencoder using the method in Algorithm 1, then obtain the abstract features extracted by the encoder as,

$$\mathbf{X}^{(i)} = \mathbf{X}\mathbf{\Gamma}^{(i)T} \tag{22}$$

After obtaining $k$ abstract features extracted by REKELM-AEs, combine all the results into a feature matrix $\mathbf{X}^{final} = [\mathbf{X}^{(1)}, \mathbf{X}^{(2)}, \cdots, \mathbf{X}^{(k)}]$, and finally use $\mathbf{X}^{final}$ as input to train an ELM classifier $\mathbf{H}^{final}\boldsymbol{\beta} = \mathbf{T}$.

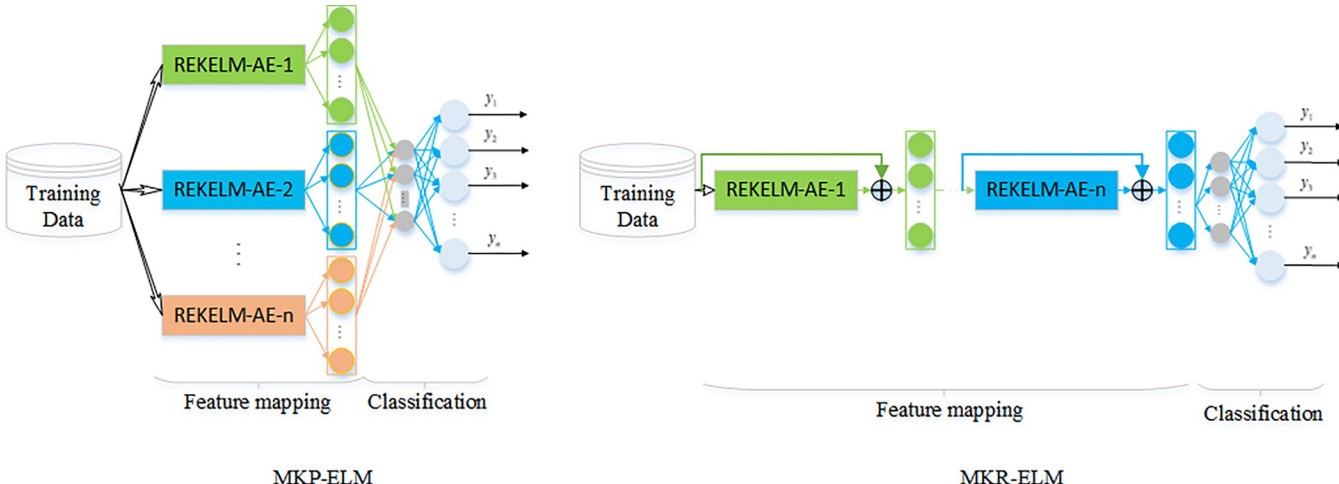

**Fig 3. Network structure of MKP-ELM and MKR-ELM.**

Residual connection has been proved to be an effective and efficient principle in deep neural networks [2,3]. In order to further exploit the effectiveness of the proposed RKELM-AE, another deep model is designed by stacking multiple REKELM-AEs. Furthermore, the residual connection is adopted between each autoencoder, as illustrated in Fig 3. The output $\mathbf{Y}^{(i)}$ of one REKELM-AE is,

$$\mathbf{Y}^{(i)} = \mathbf{X}^{input} + \mathbf{X}^{output} \tag{23}$$

Where $\mathbf{X}^{input}, \mathbf{X}^{output}$ are the input and output of the encoder. Different REKELM-AE has different kernel function.

Stacking multiple REKELM-AEs can obtain multiple kernel mappings, while residual connection eases the information loss during the transmission. In this paper, 4 different kernel functions (or similarity functions) are selected, namely radial basis function ($K_1(\cdot)$), Euclidean distance ($K_2(\cdot)$), Manhattan distance ($K_3(\cdot)$), and cosine similarity ($K_4(\cdot)$):

$$\begin{cases} K_1(\mathbf{x}_i, \mathbf{r}_j) = \exp(-\|\mathbf{x}_i - \mathbf{r}_j\|_2^2 / 2\sigma^2) \\ K_2(\mathbf{x}_i, \mathbf{r}_j) = \|\mathbf{x}_i - \mathbf{r}_j\|_2^2 \\ K_3(\mathbf{x}_i, \mathbf{r}_j) = \|\mathbf{x}_i - \mathbf{r}_j\|_1 \\ \quad K_4(\mathbf{x}_i, \mathbf{r}_j) = \mathbf{x}_i \mathbf{r}_j \Big/ \sqrt{\|\mathbf{x}_i\|_2^2 \|\mathbf{r}_j\|_2^2} \end{cases} \tag{24}$$

## 4.2 Cost-sensitive classification

For a cost-sensitive classification problem with $m$ classes, the cost matrix is:

$$\mathbf{C} = \begin{bmatrix} c_{11} & \cdots & c_{1m} \\ \vdots & \ddots & \vdots \\ c_{m1} & \cdots & c_{mm} \end{bmatrix} \tag{25}$$

Where $c_{ij}$ is the cost of classifying the $i$ class as the $j$ class. Then, for any $\mathbf{x}$, the category based

on minimum risk decision can be expressed as:

$$j* = \arg \min_{j=1,2,\cdots,m} \left( \sum_{i=1}^{m} P(i|\mathbf{x}) \cdot c_{ij} \right) \tag{26}$$

The key to implementing minimum risk decision is estimating the posterior probability. Therefore, the hard output of ELM needs to be converted into probability output. This paper adopts the method from [31] to use the Sigmoid function to convert the output of MKP-ELM into posterior probabilities:

$$P(i|\mathbf{x}) = \frac{1}{1 + \exp(-f(\mathbf{x}))} \tag{27}$$

Where $f(\mathbf{x})$ is the output of the classifier for class $\mathbf{x}$. When the number of categories is greater than 2, it cannot be guaranteed that Eq (27) will sum to 1 for all classes. Therefore, further normalization of the estimation of Eq (27) is needed:

$$\tilde{P}(i|\mathbf{x}) = \frac{P(i|\mathbf{x})}{\sum_{j=1}^{m} P(j|\mathbf{x})} \tag{28}$$

Once we obtain the estimation of unknown sample $\mathbf{x}$, we can then use the minimum risk decision criterion in Eq (26) for classification.

## 5. Experiment and analysis

This section validates the effectiveness of MKP-CSELM. The experiments consist of three parts: first, verifying the impact of different numbers of nodes in the first hidden layer ($L$) and random sampling numbers in the second hidden layer ($\hat{N}$) on the performance of the autoencoder in REKELM-AE; then validating the effectiveness of MKP-CSELM in handling cost-sensitive issues on 21 UCI datasets. Finally, further validating the performance of the proposed method on a cost-sensitive pulmonary pathology recognition dataset.

### 5.1 Implementation design

There are three evaluation matrices for classifiers: accuracy rate $acc$, total misclassification cost $Tc$, and relative performance $r_\alpha$, defined as follows:

$$acc = 1 - \frac{\sum_{i=1}^{m} err_i}{N}, \; Tc = \sum_{i=1}^{m} \sum_{j=1}^{m} err_{ij} c_{ij}, \; r_\alpha = \frac{Tc_\alpha}{\max_{i=1,2,\cdots,m} Tc_i} \tag{29}$$

Where $err_i$ is the number of misclassified samples in class $i$, $err_{ij}$ is the number of samples misclassified as class $j$ when they belong to class $i$, and $Tc_\alpha$ is the misclassification cost of algorithm $\alpha$.

In addition to MKP-CSELM, seven methods were selected for comparison, including two cost-sensitive ELM methods: cost-sensitive ELM (CELM) [24], cost-sensitive voting ELM (CSVELM) [30]; HELM [27], three cost-sensitive naïve Bayes methods: cost-sensitive naïve Bayes (CSNB) [32], cost-sensitive Bayesian network (CSBN) [33], multi-class cost-sensitive $k$-nearest neighbor (mcKNN) [34]; and one cost-sensitive neural network method: cost-sensitive neural network (CSNN) [34]. For CELM, sample weights were calculated using the method in Eq (16) based on the misclassification cost; the base classifier number for CSVELM was set to 30; the number of neighbors for mcKNN was set to 3 as in reference [34]. For CELM and CSVELM, the hidden layer activation function is both set to the Sigmoid function. The number of hidden nodes in the classifier, represented as $L$, and the regularization parameter,

**Table 1. UCI dataset.**

| Dataset | No. of feature | No. of category | No. of sample |
|---|---|---|---|
| Breast-w | 9 | 2 | 699 |
| Diabetes | 8 | 2 | 768 |
| Heart statlog | 13 | 2 | 270 |
| Ionosphere | 34 | 2 | 351 |
| Sonar | 60 | 2 | 208 |
| Balance | 4 | 3 | 625 |
| Ecoli | 7 | 8 | 366 |
| Glass | 9 | 3 | 214 |
| Iris | 4 | 3 | 150 |
| Letter | 16 | 26 | 1214 |
| Page blocks | 10 | 5 | 5473 |
| Sat | 36 | 6 | 6435 |
| Segmentation | 19 | 7 | 2310 |
| shuttle | 9 | 7 | 14500 |
| Soybean | 35 | 18 | 306 |
| Thyriod | 5 | 3 | 215 |
| Vehicle | 18 | 4 | 846 |
| Vowel | 13 | 11 | 990 |
| Wine | 13 | 3 | 178 |
| Yeast | 8 | 10 | 1484 |
| Zoo | 16 | 7 | 101 |

represented as $C$, are selected from the set $[200,400,\cdots,2000]$ and $[2^{-6},2^{-4},\cdots,2^{12}]$ respectively to find the combination that minimizes the classification cost. Apart from the random encoding layer weights with a Gaussian model of variance 1 and mean 0 in MKP-CSELM [10], random weights at other locations are generated in set $(-1,1)$ using a uniform distribution. For MKP-CSELM, the number of hidden mapping layer nodes and the number of similar mapping layers were the same for each autoencoder.

The experimental data includes 5 binary datasets and 16 multiclass datasets from UCI repository, as shown in Table 1. All features were normalized to [0,1]. Due to the large sample sizes of the Page Blocks, Sat, Segmentation, and Shuttle datasets, memory constraints were encountered during experimentation. Therefore, 1000 random samples were selected from each dataset for training and testing. For each dataset, 60% of the samples were randomly selected as the training set, and the remaining 40% were used as the test set. The experiments were conducted using MATLAB R2022a on a desktop computer with an Intel Core i9 CPU and 16GB RAM.

## 5.2 Ablation study

To examine the effectiveness of each component in our proposed framework, a series of ablation experiments are performed on UCI dataset.

**a) Evaluation on different hidden layer nodes and reference points.** The representational capability of REKELM-AE is mainly influenced by the number of random hidden layer nodes $L_1$ and the number of reference points for similar hidden layers $\bar{N}$. The first hidden layer can be seen as an extension of the original input dimensions, and the similar mapping layer can be viewed as a feature compression process on the output of the former, two

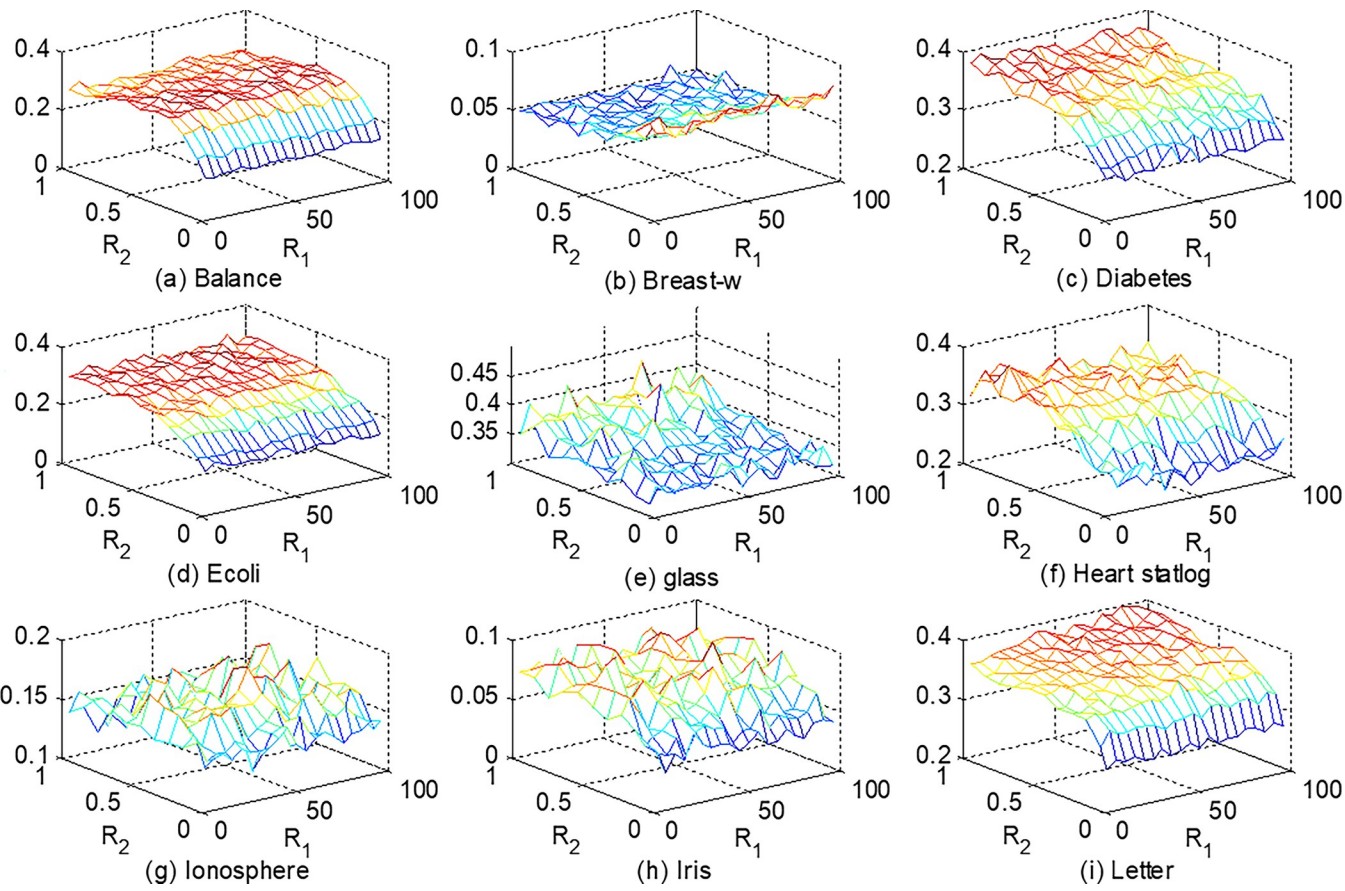

**Fig 4. The error rates of MKP-ELM on different extension and compression degrees.**

parameters are defined to measure the degree of extension and compression,

$$R_1 = L_1/d \quad R_2 = \widehat{N}/N$$

Where $d$ represents the original input feature dimension, and $N$ is the number of training samples.

This section only compares the impact of different parameters on the representational capability of REKELM-AE, hence neglecting the misclassification cost. By fixing $C = 500$, $L = 2000$ for ELM and $R_1$ varies from 5 to 100 and $R_2$ ranges from 0.1 to 1, Fig 4 shows the average error rates of MKP-ELM by running 10 times independently.

From Fig 4, it can be observed that the classification error rate of REKELM-AE does not vary significantly with different values of $R_2$, indicating that the classifier is not sensitive to the number of nodes in the first hidden layer. On the other hand, in 7 out of the 9 datasets (Balance, Diabetes, Ecoli, Glass, Heart statlog, Iris, and Letter), there is a clear downward trend in the classification error rate as $R_2$ decreases. Only in one dataset (Breast-w), the classification error rate increases as $R_2$ decreases. This suggests that an effective similarity mapping of input data can be achieved using only a small number of reference points; utilizing a larger $\widehat{N}$ (or $R_2$) and selecting more training data as reference points may yield redundant features after similarity mapping, resulting in a decrease in classification performance. Therefore, it is completely unnecessary to compute the similarity matrix using a larger portion or the entire training set.

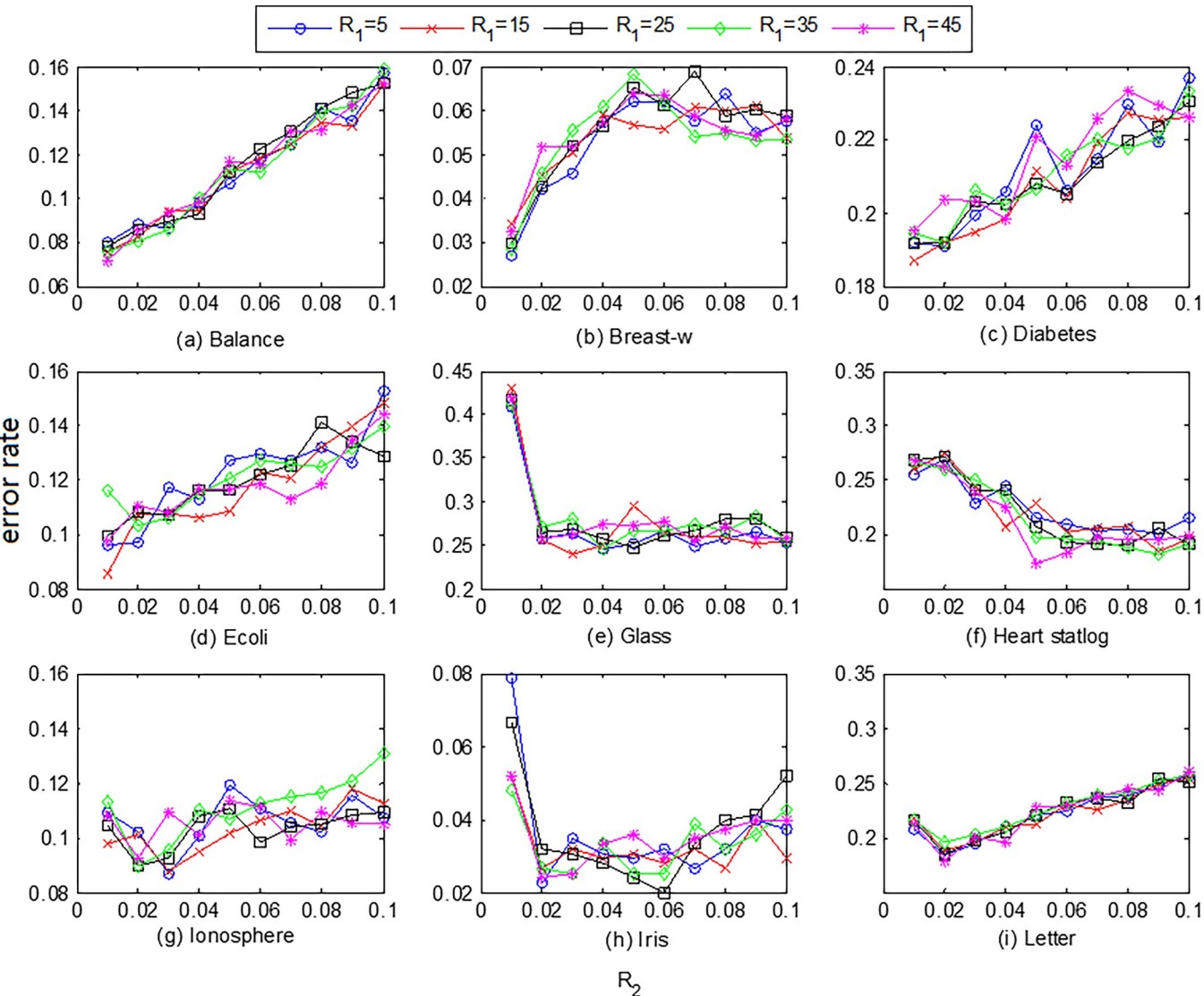

**Fig 5. Results for different values of A, ranging from 0.01 to 0.1.**

To further discuss the impact of different values of $R_2$ on classification performance, Fig 5 provides results for different values of $R_1$ while $R_2$ ranges from 0.01 to 0.1. It can be seen that the error rates of 4 out of the 9 datasets (Balance, Breast-w, Diabetes, Ecoli) further decrease as $R_2$ decreases, indicating that for these four datasets, only a very small number of reference points are needed to achieve similarity mapping. On the other hand, four datasets (Glass, Ionosphere, Iris, Letter) achieve optimal values around $R_2 = 0.02$, and as $R_2$ is further reduced, the classification error rate increases. This is mainly because when $R_2$ is very low, only one or a few reference points may be selected, which is insufficient to achieve similarity mapping of the input data.

**b) Evaluation on different models.** In this part, the effectiveness of the proposed two different models is investigated. For MKP-ELM and MKR-ELM, the kernel functions, No. of $\bar{N}$, NO. of $L_1$ are all the same. The cost matrix is randomly produced in a range of [1,20]. The average results on all datasets are presented in Table 2.

**Table 2. comparison with different model.**

| Kernel function combination | Average *acc*(%) | | Average *Tc* | |
|---|---|---|---|---|
| | MKP-ELM | MKR-ELM | MKP-ELM | MKR-ELM |
| $K_1(\cdot)$ | 83.22 | 83.26 | 9.44e1 | 9.43e1 |
| $K_1(\cdot)+K_2(\cdot)$ | 84.35 | 84.40 | 9.33e1 | 9.27e1 |
| $K_1(\cdot)+K_2(\cdot)+K_3(\cdot)$ | 84.34 | 84.41 | 9.36e1 | 9.27e1 |
| $K_1(\cdot)+K_2(\cdot)+K_3(\cdot)+K_4(\cdot)$ | 84.40 | 84.57 | 9.26e1 | 9.09e1 |

The results show that the performance of MKR-ELM is slightly better than that of MKP-ELM when there is only one kernel function, indicating that residual connection is effective for improving performance. When the kernel function is increased, the performance increase of MKP-ELM is smaller than that of MKR-ELM, indicating that firstly, increasing the kernel function can improve the performance, and secondly, vertically stacking multiple autoencoders with residual connections is more effective than simple horizontal expansion.

**c) Evaluation on different kernel combinations.** In this part, the effectiveness of different kernel function combination is investigated. In order to thoroughly analysis the influence of kernel function, we have exhausted all possible combinations under different number of kernel functions. For example, when the number of kernel functions is 2, we conduct six sets of experiments. The average results are shown in Table 3.

The results in Tables 2 and 3 both show that with the increase of the number of kernel functions, the performance of both models is improved. When the number of kernel functions is increased from 1 to 2, the performance increases significantly. Models with two and three kernel functions have similar performance, but when the number of kernel functions is increased from 3 to 4, the performance increases significantly. These indicate that different kernel functions have different feature mapping ability.

## 5.3 Results and analysis on UCI datasets

In this section, the effectiveness of MKP-CSELM is validated on 21 UCI datasets. To reduce computational complexity, the regularization factor and the numbers of hidden nodes are fixed as $C = 100$ $L = 1000$, respectively. Both the first hidden layer of the autoencoder and the hidden layer of the classifier use the Sigmoid activation function. As analyzed in Section 4.2, $R_1$ has a minor impact on the classifier, while the performance of MKP-ELM fluctuates significantly around $R_2 = 0.1$. For MKP-CSELM, keeping $R_1 = 10$ fixed, search for the value of $R_2$ within the range of [0.01,0.2] with a step size of 0.01 to minimize the classification cost of the classifier. For the cost matrix, three types (Type-a, Type-b, Type-c) are generated with a maximum value of 10 for each type. The training and testing sets are the same for all methods.

Every method runs 10 independent repetitions. Due to space limitations, only the results on Type-a is presented. Table 4 shows the average accuracy and misclassification cost. The "↓","↑" and "≈" marks imply that the result of MKP-CSELM is significantly better than, worse

**Table 3. Comparison with different kernel combination.**

| Kernel function combination | Average *acc*(%) | | Average *Tc* | |
|---|---|---|---|---|
| | MKP-ELM | MKR-ELM | MKP-ELM | MKR-ELM |
| 1 kernel | 81.43 | 81.50 | 1.06e2 | 1.03e2 |
| 2 kernels | 83.77 | 83.98 | 9.88e1 | 9.80e1 |
| 3 kernels | 83.69 | 84.11 | 9.86e1 | 9.68e1 |
| 4 kernels | 84.40 | 84.57 | 9.26e1 | 9.09e1 |

than and similar to the compared result with 0.05 significance level, respectively. In Table 4, in terms of accuracy, MKP-CSELM obtained the most optimal values (6), followed by mcKNN (4). As for the three data sets (Segmentation, Vehicle and Vowel), the accuracy of MKP-CSELM has been significantly improved compared with other methods, 3.0%, 3.3% and 7.3% higher than that of the second best method, and more than 20% higher than that of the worst method. For the misclassification cost, MKP-CSELM achieved optimal values on a total of 10 datasets, nearly half of the total number of datasets. Similarly, the misclassification cost of MKP-CSELM on Segmentation, Vehicle and Vowel data sets is significantly lower than that of other methods, indicating that the overall recognition performance of MKP-CSELM on the above three data sets has been greatly improved.

Table 5 summarizes the results of rank-sum tests under three cost matrices. Each column indicates the number of dataset which MKP-CSELM is significantly better than, worse than and similar to the compared method with 0.05 significance level. For example, the first column under Type-a, 3,10,8 indicate MKP-CSELM is significantly worse than, similar to and better than CELM on 3,10,8 datasets, respectively. The results show that under all three cost matrices, MKP-CSELM outperforms the other 7 methods significantly in both indicators on the majority of datasets. Table 6 provides the average rank of Friedman test results. Overall, it can observed that the rank of MKP-CSELM is the best for both indicators under the three cost matrices, followed by CELM. The performance of CSNB is the worst, followed by CSVELM, indicating that the multi-classifier voting decision cannot estimate posterior probabilities well. Relative performance $r_\alpha$ of the seven methods for each dataset is calculated, and the cumulative $r_\alpha$ of each method on all datasets under different cost matrices is shown in Fig 6. Each color layer corresponding to a dataset. The figure indicated that MKP-CSELM performs the best, followed by CELM, while CSNB shows the poorest average performance. These experimental results demonstrate the effectiveness of MKP-CSELM.

### 5.4 Case study

In this section, the proposed method is used for realistic cost-sensitive problems. LungHist700 [35] is dataset of histological images in pulmonary pathology. It consists of 691 images from 45 patients, with each image having a resolution of $1200 \times 1600$ pixels and stored in.jpg format. These images are captured at either 20x or 40x magnification levels and are categorized into seven classes (see Fig 6). An accompanying.csv file links each image to the associated patient ID. All patients have been anonymized, and the file includes an identifier to match images from the same patient.

All the images are resized to 300*400 pixels, 70% for training and 30% for testing. A ResNet50 is trained and the last feature map is used as the input of MKR-ELM. For simplicity, the samples are divided into two classes, normal and unnormal. The misclassification costs of the normal and unnormal are 1 and 10, respectively. The results are presented in Table 7. 'Normal' indicates MKR-ELM without cost-sensitive classification. The results show that, cost-sensitive classification results in a 4.76% reduction in classification accuracy, but a 42.18% percent reduction in misclassification costs.

### 6. Conclusion

This paper proposes two multi-kernel cost-sensitive ELM models based on the expected kernel function. Firstly, the kernel function is reinterpreted from the perspective of similarity. Based on the kernel ELM, a simplified kernel autoencoder model is presented by randomly selecting a subset of samples from the input data as reference points. According to the expected kernel ELM theory, a random mapping layer is added after the input layer to design a simplified

**Table 4. Results of Type-a.**

| Dataset | acc(%) | | | | | | | | Tc | | | | | | | |
|---|---|---|---|---|---|---|---|---|---|---|---|---|---|---|---|---|
| | CELM | CSNB | CSVELM | CSBN | mcKNN | CSNN | HELM | Ours | CELM | CSNB | CSVELM | CSBN | mcKNN | CSNN | HELM | Ours |
| Balance | 84.9↓ | 72.3↓ | **92.3**↑ | 70.2↓ | 85.1↓ | 90.9↑ | 91.7↑ | 88.1 | 3.49e1↓ | 7.88e1↓ | **2.69e1**↑ | 8.90e1↓ | 6.03e1↓ | 2.87e1≈ | 2.88e1↑ | 2.98e1 |
| Breast-w | 95.8≈ | 94.8↓ | 97.0≈ | **97.1**≈ | 95.9≈ | 94.8↓ | 96.8≈ | 97.0 | 1.21e1↑ | 1.89e1≈ | 2.21e1↓ | 1.33e1↓ | 2.69e1↓ | 2.80e1↓ | 2.32e1↓ | 1.42e1 |
| Diabetes | 69.9↓ | 75.1↑ | **76.0**↑ | 75.0↑ | 70.9↓ | 73.9↑ | 75.4↑ | 74.1 | 1.12e2≈ | 1.21e2↓ | 1.43e2↓ | 1.16e2↓ | 1.69e2↓ | 1.11e2≈ | 1.07e2↓ | **9.93e1** |
| Ecoli | 79.9↓ | 69.2↓ | **87.9**↑ | 78.1↓ | 84.7≈ | 84.0≈ | 86.8↑ | 84.2 | 3.22e1↓ | 1.56e2↓ | 5.10e1↓ | 3.88e1↓ | 3.89e1↓ | 3.86e1↓ | 4.86e1≈ | **3.11e1** |
| Glass | 49.9↓ | 60.0≈ | 61.1↓ | 54.3↓ | **67.9**↑ | 58.1≈ | 56.8↓ | 60.1 | 4.60e1↓ | 8.55e1↓ | 8.81e1↓ | 5.90e1↓ | 4.29e1↓ | 4.77e1↓ | 8.76e1↓ | **4.01e1** |
| Heart statlog | 72.4↑ | **81.1**↑ | 73.8↓ | 75.1↓ | 62.8≈ | 78.9↑ | 72.7↑ | 62.9 | 3.80e1↑ | 4.78e1≈ | 6.98e1↓ | **3.16e1**↑ | 6.00e1↓ | 6.53e1↓ | 7.01e1≈ | 4.66e1 |
| Ionosphere | 87.9≈ | 68.4↓ | 79.2↓ | 88.0≈ | 84.1↓ | 87.8≈ | 80.1↓ | **88.8** | 1.89e1↓ | 4.88e1↓ | 3.29e1↓ | 6.56e1↓ | 3.33e1↓ | 2.28e1≈ | 3.33e1↓ | **1.59e1** |
| Iris | 95.0≈ | 94.0≈ | 95.0≈ | 90.2↓ | 95.9↓ | 95.9↓ | 91.1↓ | **96.9** | 3.49e0≈ | **1.47e0**↑ | 1.44e1↓ | 1.02e1↓ | 5.56e0↓ | 5.77e0↓ | 1.31e1↓ | 1.58e0 |
| Letter | **76.0**≈ | 63.4↓ | 52.3↓ | 60.1↓ | 64.9↓ | 53.0↓ | 61.3↓ | 75.2 | **1.28e2**≈ | 1.99e2↓ | 2.70e2↓ | 2.09e2↓ | 2.08e2↓ | 2.41e2↓ | 2.61e2↓ | 1.29e2 |
| Page blocks | 93.1↑ | 85.0↓ | 90.9↓ | 91.1↓ | 94.0↓ | **95.2**↑ | 91.1↓ | 92.9 | 9.91e1≈ | 8.22e1↓ | 1.61e2↓ | 6.79e1↓ | 5.11e1↓ | **3.14e1**↑ | 1.55e2↓ | 9.80e1 |
| Sat | 79.9↓ | 79.1↓ | 82.8≈ | 81.0↓ | **87.0**↑ | 72.3↓ | 83.1≈ | 82.9 | 1.05e2≈ | 1.59e2↓ | 2.68e2↓ | 1.47e2↓ | 1.28e2↓ | 1.36e2↓ | 2.72e2↓ | **9.79e1** |
| Segmentation | 82.1↓ | 71.2↓ | 70.7↓ | 86.1↓ | 88.9↓ | 82.1↓ | 77.6↓ | **91.9** | 8.08e1↓ | 1.52e2↓ | 3.29e2↓ | 8.40e1↓ | 8.12e1↓ | 1.21e2↓ | 3.01e2↓ | **3.39e1** |
| shuttle | 84.8↓ | 81.1↓ | 79.1↓ | 95.9≈ | **97.8**↓ | 94.1≈ | 81.2↓ | 95.9 | 6.00e1↓ | 7.80e1↓ | 8.49e1↓ | 1.48e1≈ | **5.29e0**↓ | 2.58e1≈ | 7.70e1↓ | 1.59e1 |
| Sonar | 67.1≈ | 69.9≈ | 75.0≈ | 63.9↓ | 77.6↓ | 76.7≈ | 73.1≈ | 73.9 | 3.58e1↓ | 1.31e2↓ | 3.22e1≈ | 9.40e1↓ | 5.39e1↓ | 5.50e1↓ | 3.18e1≈ | **3.11e1** |
| Soybean | 89.9↓ | 60.1↓ | **89.9**≈ | 60.3↓ | 84.4↓ | 80.1↓ | 88.1≈ | 87.9 | **1.19e1**↑ | 2.22e2↓ | **1.30e1**≈ | 4.90e1↓ | 2.30e1↓ | 2.69e1↓ | 1.55e1↓ | 1.69e1 |
| Thyriod | 92.1↓ | **97.1**↑ | 85.9↓ | 91.8↓ | 91.9↓ | 95.8≈ | 87.1↓ | 94.7 | 1.46e1↓ | 5.39e0↓ | 2.70e1↓ | 1.18e1↓ | 1.20e1↓ | **4.46e0**↓ | 2.56e1↓ | 6.69e0 |
| Vehicle | 77.8≈ | 52.1↓ | 64.2↓ | 63.1↓ | 64.2↓ | 64.3↓ | 73.1↓ | **81.1** | 8.55e1↓ | 3.39e2↓ | 1.68e2↓ | 1.90e2↓ | 1.68e2↓ | 1.31e2↓ | 1.45e2↓ | **6.77e1** |
| Vowel | 70.1↓ | 61.8↓ | 40.2↓ | 59.2↓ | 89.1↓ | 45.1↓ | 78.4↓ | **96.4** | 1.33e2↓ | 1.61e2↓ | 3.38e2↓ | 1.76e2↓ | 6.82e1↓ | 2.69e2↓ | 3.22e2↓ | **1.90e1** |
| Wine | 93.1≈ | 97.0↑ | 70.1↓ | 96.3↓ | 67.9↓ | **96.0**↓ | 81.1↓ | 92.0 | 5.08e0≈ | 1.15e1≈ | 3.41e1↓ | 5.05e0≈ | 3.31e1↓ | **3.11e0**↑ | 3.28e1↓ | 6.48e0 |
| Yeast | **58.9**≈ | 47.1↓ | 54.1↓ | 55.1↓ | 53.9↓ | 57.3≈ | 55.2↓ | 57.9 | **2.42e2**≈ | 5.59e2↓ | 3.11e2≈ | 2.80e2↓ | 2.91e2≈ | 2.60e2≈ | 3.03e2↓ | 2.48e2 |
| Zoo | 95.3↓ | 42.6↓ | 93.2↓ | 83.1↓ | 87.8↓ | 91.9↓ | 90.8↓ | **95.9** | 2.00e0↓ | 4.88e1↓ | 2.79e0↓ | 6.77e0↓ | 4.70e0↓ | 3.39e0↓ | 2.67e0↓ | **1.38e0** |

**Table 5. Results of rank-sum test.**

| Cost type | | Results of rank-sum test on classification accuracy | | | | | | | Results of rank-sum test on classification cost | | | | | | |
|---|---|---|---|---|---|---|---|---|---|---|---|---|---|---|---|
| | | CELM | CSNB | CSVELM | CSBN | mcKNN | CSNN | HELM | CELM | CSNB | CSVELM | CSBN | mcKNN | CSNN | HELM |
| Type-a | ↑ | 3 | 4 | 5 | 3 | 2 | 5 | 4 | 2 | 1 | 1 | 2 | 1 | 2 | 2 |
| | ≈ | 10 | 3 | 7 | 3 | 6 | 7 | 6 | 12 | 5 | 3 | 4 | 4 | 9 | 3 |
| | ↓ | **8** | **14** | **9** | **15** | **13** | **9** | **11** | **7** | **15** | **17** | **15** | **16** | **10** | **16** |
| Type-b | ↑ | 5 | 3 | 5 | 5 | 2 | 4 | 7 | 5 | 4 | 3 | 4 | 3 | 2 | 4 |
| | ≈ | 8 | 7 | 6 | 6 | 7 | 6 | 4 | 8 | 3 | 5 | 5 | 6 | 7 | 6 |
| | ↓ | **8** | **11** | **10** | **10** | **12** | **11** | **10** | **8** | **14** | **13** | **12** | **12** | **12** | **11** |
| Type-c | ↑ | 3 | 4 | 4 | 5 | 5 | 6 | 2 | 5 | 3 | 3 | 2 | 3 | 2 | 2 |
| | ≈ | 7 | 3 | 5 | 3 | 4 | 5 | 6 | 9 | 5 | 6 | 6 | 6 | 6 | 4 |
| | ↓ | **11** | **14** | **12** | **13** | **12** | **10** | **13** | **7** | **13** | **12** | **13** | **12** | **13** | **15** |

**Table 6. Results of Friedman test.**

| Cost type | Everage rank on classification accuracy | | | | | | | |
|---|---|---|---|---|---|---|---|---|
| | CELM | CSNB | CSVELM | CSBN | mcKNN | CSNN | HELM | Ours |
| Type-a | 4.19 | 5.42 | 4.47 | 5 | 3.90 | 4.10 | 4.19 | **2.81** |
| Type-b | 4.22 | 5.38 | 5.01 | 4.96 | 4.12 | 4.36 | 3.98 | **2.99** |
| Type-c | 4.30 | 5.76 | 4.83 | 5.11 | 4.31 | 4.04 | 4.13 | **2.83** |
| | Everage rank on classification cost | | | | | | | |
| Type-a | 3.10 | 5.52 | 6.14 | 4.67 | 4.48 | 4.14 | 5.47 | **2.14** |
| Type-b | 3.38 | 5.44 | 5.50 | 4.71 | 4.61 | 4.38 | 5.82 | **2.44** |
| Type-c | 3.73 | 5.76 | 6.03 | 4.88 | 4.72 | 4.93 | 5.56 | **2.33** |

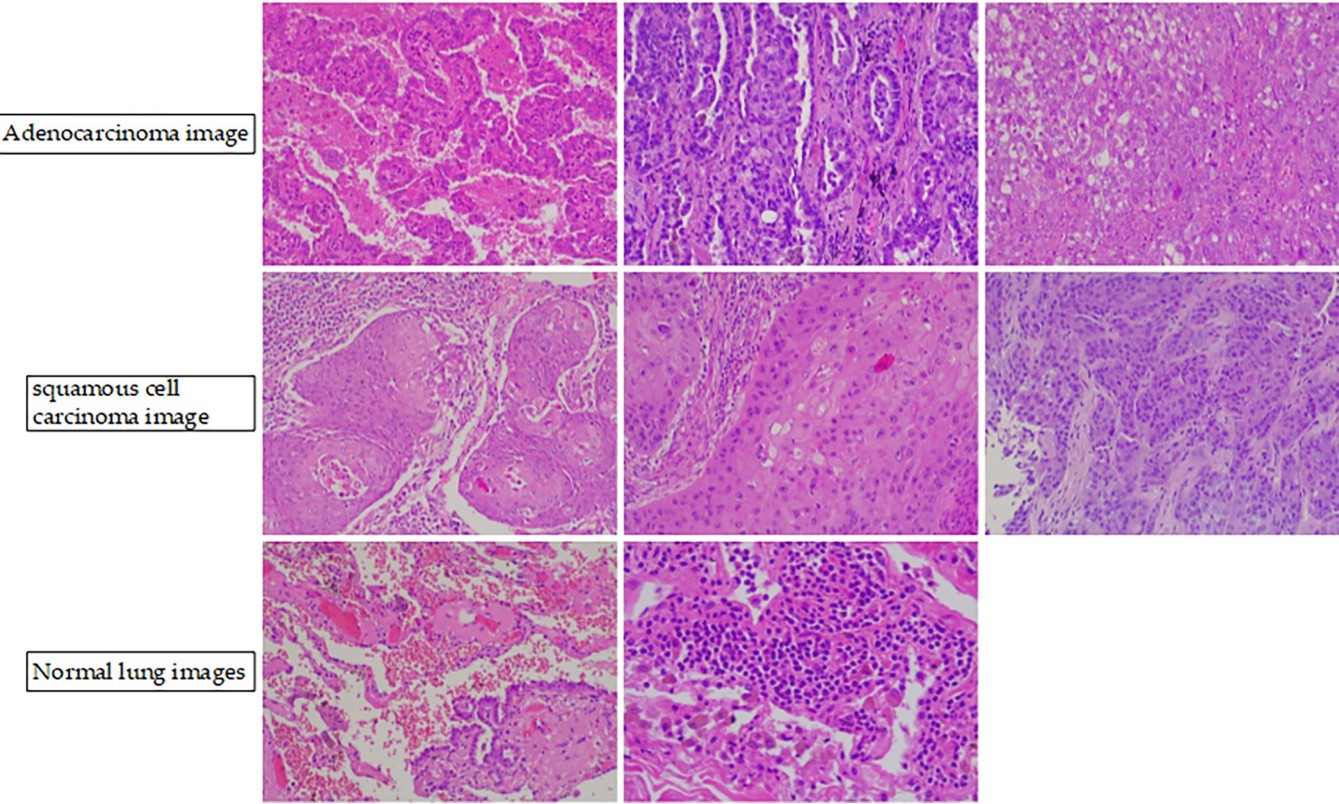

**Fig 6. Example images of LungHist700.**

**Table 7. Results on LungHist700.**

| methods | Average *acc* (%) | | Average *Tc* | |
|---|---|---|---|---|
| | Normal | Cost-sensitive | Normal | Cost-sensitive |
| results | 81.56 | 76.8 | 1420 | 821 |

expected kernel autoencoder, which effectively combines random mapping and similarity mapping. Four types of similarity kernel functions are defined, and two multi-kernel ELM models are designed using the simplified expected kernel autoencoder. The classifier output is then transformed into posterior probabilities, and cost-sensitive decisions are made based on the minimum risk criterion. Comparative analysis with six cost-sensitive methods on 21 UCI datasets shows that the proposed method can achieve better generalization performance with only a few reference points selected than the comparative methods. The case study on realistic pulmonary pathology classification further demonstrated the effectiveness of the proposed approach.

## Author Contributions

**Conceptualization:** Liang Yixuan.

**Data curation:** Liang Yixuan.

**Formal analysis:** Liang Yixuan.

**Funding acquisition:** Liang Yixuan.

**Investigation:** Liang Yixuan.

**Methodology:** Liang Yixuan.

**Project administration:** Liang Yixuan.

**Resources:** Liang Yixuan.

**Software:** Liang Yixuan.

**Supervision:** Liang Yixuan.

**Validation:** Liang Yixuan.

**Visualization:** Liang Yixuan.

**Writing – original draft:** Liang Yixuan.

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
