## [Decision Letter · Decision Letter 0]

22 Jul 2024

PONE-D-24-21789Cost-sensitive Multi-kernel ELM based on Reduced Expectation Kernel Auto-encoderPLOS ONE

Dear Dr. Liang,

Thank you for submitting your manuscript to PLOS ONE. After careful consideration, we feel that it has merit but does not fully meet PLOS ONE’s publication criteria as it currently stands. Therefore, we invite you to submit a revised version of the manuscript that addresses the points raised during the review process.

**ACADEMIC EDITOR: **

Dear Authors,

please revise proposed manuscript thoroughly according to all reviewers' comments.

Additionally, please do the following:

- Visualization of obtained results must be improved

- Motivation behind proposed research should be more clearly explain. Please elaborate what is "beyond state-of-the-art" of proposed. study.

- Make sure that the source code is available according to PLOS ONE publication policies.

- Make sure that you have conducted rigid statistical analysis.

All the best,

Nebojsa Bacanin

We look forward to receiving your revised manuscript.

Kind regards,

Nebojsa Bacanin

Academic Editor

PLOS ONE

Additional Editor Comments:

Dear Authors,

please revise proposed manuscript thoroughly according to all reviewers' comments.

Additionally, please do the following:

- Visualization of obtained results must be improved

- Motivation behind proposed research should be more clearly explain. Please elaborate what is "beyond state-of-the-art" of proposed. study.

- Make sure that the source code is available according to PLOS ONE publication policies.

- Make sure that you have conducted rigid statistical analysis.

All the best,

Nebojsa Bacanin

Reviewers' comments:

Reviewer's Responses to Questions

**Comments to the Author**

1. Is the manuscript technically sound, and do the data support the conclusions?

Reviewer #1: Yes

Reviewer #2: Yes

Reviewer #3: Partly

2. Has the statistical analysis been performed appropriately and rigorously? 

Reviewer #1: Yes

Reviewer #2: Yes

Reviewer #3: Yes

3. Have the authors made all data underlying the findings in their manuscript fully available?

Reviewer #1: Yes

Reviewer #2: Yes

Reviewer #3: Yes

4. Is the manuscript presented in an intelligible fashion and written in standard English?

Reviewer #1: Yes

Reviewer #2: Yes

Reviewer #3: Yes

5. Review Comments to the Author

Reviewer #1: The manuscript presents a novel approach to enhance the performance of Extreme Learning Machines (ELM) through a cost-sensitive multi-kernel method based on a reduced expectation kernel auto-encoder. The authors aim to address the issues of long training times and the complexity of setting kernel function weights in existing multi-kernel models.

Strengths:

1. The introduction of a reduced expectation kernel auto-encoder to design a multi-kernel ELM model is innovative.

2. The manuscript is well-written and the methodology is clearly explained.

3. The experimental results are comprehensive and validate the effectiveness of the proposed method.

Weaknesses:

1. The manuscript lacks a detailed comparative analysis with other state-of-the-art methods.

2. The practical applicability of the proposed method in real-world scenarios is not sufficiently discussed.

3. The sensitivity of the model to different parameters (e.g., number of reference points) should be explored in more detail.

Major Comments:

1. Include a more detailed comparative analysis with recent state-of-the-art methods in the field like below to highlight the advantages and limitations of the proposed approach.

Speech Emotion Recognition Using Deep Sparse Auto-Encoder Extreme Learning Machine with a New Weighting Scheme and Spectro-Temporal Features Along with Classical Feature …

Speech Emotion Recognition Using Multi-Layer Sparse Auto-Encoder Extreme Learning Machine and Spectral/Spectro-Temporal Features with New Weighting Method for Data Imbalance

2. Perform a sensitivity analysis on key parameters such as the number of reference points and the choice of kernel functions. This will help in understanding the robustness of the proposed model.

3. Discuss the potential real-world applications of the proposed method and how it can be integrated into existing systems. Providing a case study or example would strengthen the manuscript.

Minor Comments:

1. Expand the literature review to include more recent works related to cost-sensitive learning and multi-kernel ELMs.

2. Including pseudocode for the proposed algorithm could enhance understanding and reproducibility.

3. Ensure that all figures and tables are clearly labeled and referenced appropriately in the text. Some figures could benefit from more detailed captions.

Reviewer #2: The proposed paper is dealing with multi-kernel parallel cost-sensitive ELM model.

I have following comments to the author:

1. Some of the references are very old, like [1], [7], I would recommend some of the following more recent papers dealing with swarm intelligence optimization of elm:

N. Bacanin, M. Zivkovic, M. Antonijevic, K. Venkatachalam, J. Lee, Y. Nam, M. Marjanovic, I. Strumberger, M. Abouhawwash, Addressing feature selection and extreme learning machine tuning by diversity-oriented social network search: an application for phishing websites detection, Complex & Intelligent Systems, pp. 1 - 36, Jun, 2023

Laifi, A., Benmohamed, E. & Ltifi, H. Xavier-PSO-ELM-based EEG signal classification method for predicting epileptic seizures. Multimed Tools Appl 83, 30675–30696 (2024). https://doi.org/10.1007/s11042-023-16514-3

Wang F, Liang Y, Lin Z, Zhou J, Zhou T. SSA-ELM: A Hybrid Learning Model for Short-Term Traffic Flow Forecasting. Mathematics. 2024; 12(12):1895. https://doi.org/10.3390/math12121895

N. Bacanin, C. Stoean, M. Zivkovic, D. Jovanovic, M. Antonijevic, D. Mladenovic, Multi-Swarm Algorithm for Extreme Learning Machine Optimization, SENSORS, Vol. 22, No. 11, pp. 1 - 34, May, 2022

2. The author has used 60% of dataset for training and 40% for test set. This is not a common ratio. Can you elaborate on the reason for this exact split ratio?

3. I do not understand Table 2. Can you please clarify in the text (give an example) how to interpret the presented results in the Table 2?

The paper is overall well written.

Reviewer #3: Cost sensitive learning is a hot topic in data mining field. Extreme learning machine (ELM), as a simple and efficient machine learning algorithm, still has some shortcomings in processing tasks related to time series data. Aiming at the problems of low training efficiency and difficult weight setting for kernel ELM autoencoder, the author proposes a multi-core parallel ELM method based on expectation kernel autoencoder, which has certain theoretical significance and application value. This paper reviews the specific application scenarios of ELM and its variants in cost-sensitive learning tasks. The following innovations have been achieved:

1) Firstly, from the point of view of calculating similarity, the reduced kernel autoencoder is defined by randomly selecting reference points from the input data;

2) Secondly, based on expected kernel ELM, a simplified expected kernel autoencoder is designed to realize the combination of random mapping and similar mapping;

3) Finally, cost-sensitive learning is implemented based on the minimum risk criterion.

The structure of the manuscript is reasonable and the discussion is clear. However, the following questions remain:

1) A large number of formula definitions have been given in relevant work, but there are few literature summaries on the progress of ELM in cost-sensitive learning tasks, which needs to be further condensed;

2) Some formulas in the paper are ambiguous, and the characters of formula (19) are problematic;

3) In the fourth part of the manuscript, the feature mapping is carried out by combining multiple ELM. The expression of how the output k feature values correspond to m hidden layer vectors is not clear and lacks innovation;

4) Detailed introduction of baseline methods and ablation experiments were lacking in comparison experiments

In summary, this manuscript is not recommended for acceptance

6. PLOS authors have the option to publish the peer review history of their article (what does this mean?). If published, this will include your full peer review and any attached files.

Reviewer #1: No

Reviewer #2: No

Reviewer #3: No

---

## [Author Response · Author response to Decision Letter 0]

12 Nov 2024

Dear reviewers and editor:

Thank you very much for your valuable suggestions and comments. All the comments are revised seriously. We hope that the revision is acceptable, and we look forward to hearing from you soon.

---

## [Decision Letter · Decision Letter 1]

19 Nov 2024

Cost-sensitive Multi-kernel ELM based on Reduced Expectation Kernel Auto-encoder

PONE-D-24-21789R1

Dear Dr. Liang,

We’re pleased to inform you that your manuscript has been judged scientifically suitable for publication and will be formally accepted for publication once it meets all outstanding technical requirements.

Kind regards,

Longhui Qin, Ph.D.

Academic Editor

PLOS ONE

Additional Editor Comments (optional):

Reviewers' comments:

Reviewer's Responses to Questions

**Comments to the Author**

1. If the authors have adequately addressed your comments raised in a previous round of review and you feel that this manuscript is now acceptable for publication, you may indicate that here to bypass the “Comments to the Author” section, enter your conflict of interest statement in the “Confidential to Editor” section, and submit your "Accept" recommendation.

Reviewer #1: (No Response)

Reviewer #2: All comments have been addressed

Reviewer #3: All comments have been addressed

2. Is the manuscript technically sound, and do the data support the conclusions?

Reviewer #1: (No Response)

Reviewer #2: Yes

Reviewer #3: Yes

3. Has the statistical analysis been performed appropriately and rigorously? 

Reviewer #1: (No Response)

Reviewer #2: Yes

Reviewer #3: Yes

4. Have the authors made all data underlying the findings in their manuscript fully available?

Reviewer #1: (No Response)

Reviewer #2: Yes

Reviewer #3: Yes

5. Is the manuscript presented in an intelligible fashion and written in standard English?

Reviewer #1: (No Response)

Reviewer #2: Yes

Reviewer #3: Yes

6. Review Comments to the Author

Reviewer #1: All the mentioned comments have been applied by the authores completely. The paper should be accepted.

Reviewer #2: (No Response)

Reviewer #3: The author has satisfactorily addressed the reviewers' concerns, and the revised manuscript now aligns with the publication standards. Consequently, I am pleased to accept it for publication.

7. PLOS authors have the option to publish the peer review history of their article (what does this mean?). If published, this will include your full peer review and any attached files.

Reviewer #1: No

Reviewer #2: No

Reviewer #3: No

---

## [Editor Report · Acceptance letter]

26 Nov 2024

PONE-D-24-21789R1 

PLOS ONE

Dear Dr. Yixuan, 

I'm pleased to inform you that your manuscript has been deemed suitable for publication in PLOS ONE. Congratulations! Your manuscript is now being handed over to our production team.

Kind regards, 

on behalf of

Prof. Longhui Qin 

Academic Editor

PLOS ONE